# Rethinking Kernel Methods for Node Representation Learning on Graphs

**Yu Tian**[*]
Rutgers University
yt219@cs.rutgers.edu

**Long Zhao**[*]
Rutgers University
lz311@cs.rutgers.edu

**Xi Peng**
University of Delaware
xipeng@udel.edu

**Dimitris N. Metaxas**
Rutgers University
dnm@cs.rutgers.edu

## Abstract

Graph kernels are kernel methods measuring graph similarity and serve as a standard tool for graph classification. However, the use of kernel methods for node classification, which is a related problem to graph representation learning, is still ill-posed and the state-of-the-art methods are heavily based on heuristics. Here, we present a novel theoretical kernel-based framework for node classification that can bridge the gap between these two representation learning problems on graphs. Our approach is motivated by graph kernel methodology but extended to learn the node representations capturing the structural information in a graph. We theoretically show that our formulation is as powerful as any positive semidefinite kernels. To efficiently learn the kernel, we propose a novel mechanism for node feature aggregation and a data-driven similarity metric employed during the training phase. More importantly, our framework is flexible and complementary to other graph-based deep learning models, e.g., Graph Convolutional Networks (GCNs). We empirically evaluate our approach on a number of standard node classification benchmarks, and demonstrate that our model sets the new state of the art. The source code is publicly available at `https://github.com/bluer555/KernelGCN`.

## 1   Introduction

Graph structured data, such as citation networks [11, 22, 30], biological models [12, 45], grid-like data [36, 37, 51] and skeleton-based motion systems [6, 42, 49, 50], are abundant in the real world. Therefore, learning to understand graphs is a crucial problem in machine learning. Previous studies in the literature generally fall into two main categories: (1) *graph classification* [8, 19, 40, 47, 48], where the whole structure of graphs is captured for similarity comparison; (2) *node classification* [1, 19, 38, 41, 46], where the structural identity of nodes is determined for representation learning.

For graph classification, kernel methods, i.e., graph kernels, have become a standard tool [20]. Given a large collection of graphs, possibly with node and edge attributes, such algorithms aim to learn a kernel function that best captures the similarity between any two graphs. The graph kernel function can be utilized to classify graphs via standard kernel methods such as support vector machines or $k$-nearest neighbors. Moreover, recent studies [40, 47] also demonstrate that there has been a close connection between Graph Neural Networks (GNNs) and the Weisfeiler-Lehman graph kernel [32], and relate GNNs to the classic graph kernel methods for graph classification.

---

[*]indicates equal contributions.

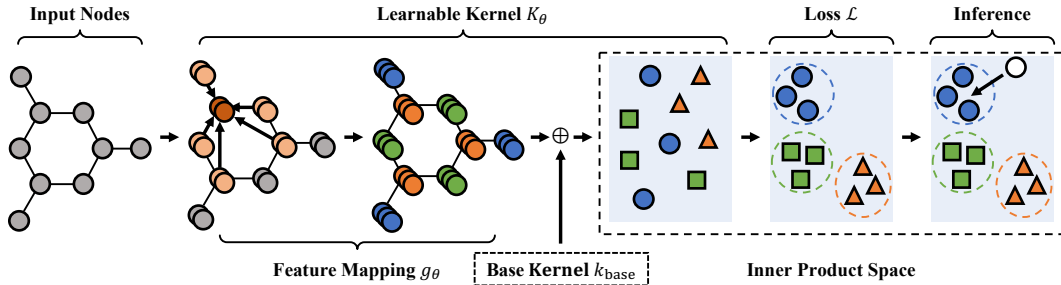

Figure 1: Overview of our kernel-based framework.

Node classification, on the other hand, is still an ill-posed problem in representation learning on graphs. Although identification of node classes often leverages their features, a more challenging and important scenario is to incorporate the graph structure for classification. Recent efforts in Graph Convolutional Networks (GCNs) [19] have made great progress on node classification. In particular, these efforts broadly follow a recursive neighborhood aggregation scheme to capture structural information, where each node aggregates feature vectors of its neighbors to compute its new features [1, 41, 46]. Empirically, these GCNs have achieved the state-of-the-art performance on node classification. However, the design of new GCNs is mostly based on empirical intuition, heuristics, and experimental trial-and-error.

In this paper, we propose a novel theoretical framework leveraging kernel methods for node classification. Motivated by graph kernels, our key idea is to decouple the kernel function so that it can be learned driven by the node class labels on the graph. Meanwhile, its validity and expressive power are guaranteed. To be specific, this paper makes the following *contributions*:

- We propose a learnable kernel-based framework for node classification. The kernel function is decoupled into a feature mapping function and a base kernel to ensure that it is valid as well as learnable. Then we present a data-driven similarity metric and its corresponding learning criteria for efficient kernel training. The implementation of each component is extensively discussed. An overview of our framework is shown in Fig. 1.

- We demonstrate the validity of our learnable kernel function. More importantly, we theoretically show that our formulation is powerful enough to express any valid positive semidefinite kernels.

- A novel feature aggregation mechanism for learning node representations is derived from the perspective of kernel smoothing. Compared with GCNs, our model captures the structural information of a node by aggregation in a single step, other than a recursive manner, thus is more efficient.

- We discuss the close connection between the proposed approach and GCNs. We also show that our method is flexible and complementary to GCNs and their variants but more powerful, and can be leveraged as a general framework for future work.

## 2 Related Work

**Graph Kernels.** Graph kernels are kernels defined on graphs to capture the graph similarity, which can be used in kernel methods for graph classification. Many graph kernels are instances of the family of convolutional kernels [15]. Some of them measure the similarity between walks or paths on graphs [4, 39]. Other popular kernels are designed based on limited-sized substructures [18, 33, 31, 32]. Most graph kernels are employed in models which have learnable components, but the kernels themselves are hand-crafted and motivated by graph theory. Some learnable graph kernels have been proposed recently, such as Deep Graph Kernels [43] and Graph Matching Networks [21]. Compared to these approaches, our method targets at learning kernels for node representation learning.

**Node Representation Learning.** Conventional methods for learning node representations largely focus on matrix factorization. They directly adopt classic techniques for dimension reduction [2, 3]. Other methods are derived from the random walk algorithm [23, 26] or sub-graph structures [13, 35, 44, 28]. Recently, Graph Convolutional Networks (GCNs) have emerged as an effective class of

models for learning representations of graph structured data. They were introduced in [19], which consist of an iterative process aggregating and transforming representation vectors of its neighboring nodes to capture structural information. Recently, several variants have been proposed, which employ self-attention mechanism [38] or improve network architectures [41, 46] to boost the performance. However, most of them are based on empirical intuition and heuristics.

# 3 Preliminaries

We begin by summarizing some of the most important concepts about kernel methods as well as representation learning on graphs and, along the way, introduce our notations.

**Kernel Concepts.** A kernel $K : \mathcal{X} \times \mathcal{X} \mapsto \mathbb{R}$ is a function of two arguments: $K(x, y)$ for $x, y \in \mathcal{X}$. The kernel function $K$ is symmetric, i.e., $K(x, y) = K(y, x)$, which means it can be interpreted as a measure of similarity. If the Gram matrix $\mathbf{K} \in \mathbb{R}^{N \times N}$ defined by $\mathbf{K}(i, j) = K(x_i, x_j)$ for any $\{x_i\}_{i=1}^N$ is *positive semidefinite (p.s.d.)*, then $K$ is a p.s.d. kernel [24]. If $K(x, y)$ can be represented as $\langle \Psi(x), \Psi(y) \rangle$, where $\Psi : \mathcal{X} \mapsto \mathbb{R}^D$ is a feature mapping function, then $K$ is a valid kernel.

**Graph Kernels.** In the graph space $\mathcal{G}$, we denote a graph as $G = (V, E)$, where $V$ is the set of nodes and $E$ is the edge set of $G$. Given two graphs $G_i = (V_i, E_i)$ and $G_j = (V_j, E_j)$ in $\mathcal{G}$, the graph kernel $K_G(G_i, G_j)$ measures the similarity between them. According to the definition in [29], the kernel $K_G$ must be p.s.d. and symmetric. The graph kernel $K_G$ between $G_i$ and $G_j$ is defined as:

$$K_G(G_i, G_j) = \sum_{v_i \in V_i} \sum_{v_j \in V_j} k_{\text{base}}(f(v_i), f(v_j)), \tag{1}$$

where $k_{\text{base}}$ is the base kernel for any pair of nodes in $G_i$ and $G_j$, and $f : V \mapsto \Omega$ is a function to compute the feature vector associated with each node. However, deriving a new p.s.d. graph kernel is a non-trivial task. Previous methods often implement $k_{\text{base}}$ and $f$ as the dot product between hand-crafted graph heuristics [25, 31, 4]. There are little learnable parameters in these approaches.

**Representation Learning on Graphs.** Although graph kernels have been applied to a wide range of applications, most of them depend on hand-crafted heuristics. In contrast, representation learning aims to automatically learn to encode graph structures into low-dimensional embeddings. Formally, given a graph $G = (V, E)$, we follow [14] to define representation learning as an encoder-decoder framework, where we minimize the empirical loss $\mathcal{L}$ over a set of training node pairs $\mathcal{D} \subseteq V \times V$:

$$\mathcal{L} = \sum_{(v_i, v_j) \in \mathcal{D}} \ell(\text{ENC-DEC}(v_i, v_j), s_G(v_i, v_j)). \tag{2}$$

Equation (2) has three methodological components: ENC-DEC, $s_G$ and $\ell$. Most of the previous methods on representation learning can be distinguished by how these components are defined. The detailed meaning of each component is explained as follows.

- ENC-DEC : $V \times V \mapsto \mathbb{R}$ is an *encoder-decoder function*. It contains an encoder which projects each node into a $M$-dimensional vector to generate the node embedding. This function contains a number of trainable parameters to be optimized during the training phase. It also includes a decoder function, which reconstructs pairwise similarity measurements from the node embeddings generated by the encoder.

- $s_G$ is a *pairwise similarity function* defined over the graph $G$. This function is user-specified, and it is used for measuring the similarity between nodes in $G$.

- $\ell : \mathbb{R} \times \mathbb{R} \mapsto \mathbb{R}$ is a *loss function*, which is leveraged to train the model. This function evaluates the quality of the pairwise reconstruction between the estimated value $\text{ENC-DEC}(v_i, v_j)$ and the true value $s_G(v_i, v_j)$.

# 4 Proposed Method: Learning Kernels for Node Representation

Given a graph $G$, as we can see from Eq. (2), the encoder-decoder ENC-DEC aims to approximate the pairwise similarity function $s_G$, which leads to a natural intuition: we can replace ENC-DEC with a kernel function $K_\theta$ parameterized by $\theta$ to measure the similarity between nodes in $G$, i.e.,

$$\mathcal{L} = \sum_{(v_i, v_j) \in \mathcal{D}} \ell(K_\theta(v_i, v_j), s_G(v_i, v_j)). \tag{3}$$

However, there exist two technical challenges: (1) designing a valid p.s.d. kernel which captures the node feature is non-trivial; (2) it is impossible to handcraft a unified kernel to handle all possible graphs with different characteristics [27]. To tackle these issues, we introduce a novel formulation to replace $K_\theta$. Inspired by the graph kernel as defined in Eq. (1) and the mapping kernel framework [34], our key idea is to decouple $K_\theta$ into two components: a base kernel $k_{base}$ which is p.s.d. to maintain the validity, and a learnable feature mapping function $g_\theta$ to ensure the flexibility of the resulting kernel. Therefore, we rewrite Eq. (3) by $K_\theta(v_i, v_j) = k_{base}(g_\theta(v_i), g_\theta(v_j))$ for $v_i, v_j \in V$ of the graph $G$ to optimize the following objective:

$$\mathcal{L} = \sum_{(v_i, v_j) \in \mathcal{D}} \ell(k_{base}(g_\theta(v_i), g_\theta(v_j)), s_G(v_i, v_j)). \tag{4}$$

Theorem 1 demonstrates that the proposed formulation, i.e., $K_\theta(v_i, v_j) = k_{base}(g_\theta(v_i), g_\theta(v_j))$, is still a valid p.s.d. kernel for any feature mapping function $g_\theta$ parameterized by $\theta$.

**Theorem 1.** *Let $g_\theta : V \mapsto \mathbb{R}^M$ be a function which maps nodes (or their corresponding features) to a M-dimensional Euclidean space. Let $k_{base} : \mathbb{R}^M \times \mathbb{R}^M \mapsto \mathbb{R}$ be any valid p.s.d. kernel. Then, $K_\theta(v_i, v_j) = k_{base}(g_\theta(v_i), g_\theta(v_j))$ is a valid p.s.d. kernel.*

*Proof.* Let $\Phi$ be the corresponding feature mapping function of the p.s.d. kernel $k_{base}$. Then, we have $k_{base}(z_i, z_j) = \langle \Phi(z_i), \Phi(z_j) \rangle$, where $z_i, z_j \in \mathbb{R}^M$. Substitute $g_\theta(v_i), g_\theta(v_j)$ for $z_i, z_j$, and we have $k_{base}(g_\theta(v_i), g_\theta(v_j)) = \langle \Phi(g_\theta(v_i)), \Phi(g_\theta(v_j)) \rangle$. Write the new feature mapping $\Psi(v)$ as $\Psi(v) = \Phi(g_\theta(v))$, and we immediately have that $k_{base}(g_\theta(v_i), g_\theta(v_j)) = \langle \Psi(v_i), \Psi(v_j) \rangle$. Hence, $k_{base}(g_\theta(v_i), g_\theta(v_j))$ is a valid p.s.d. kernel. $\square$

A natural follow-up question is whether our proposed formulation, in principle, is powerful enough to express any valid p.s.d. kernels? Our answer, in Theorem 2, is yes: if the base kernel has an invertible feature mapping function, then the resulting kernel is able to model any valid p.s.d. kernels.

**Theorem 2.** *Let $K(v_i, v_j)$ be any valid p.s.d. kernel for node pairs $(v_i, v_j) \in V \times V$. Let $k_{base} : \mathbb{R}^M \times \mathbb{R}^M \mapsto \mathbb{R}$ be a p.s.d. kernel which has an invertible feature mapping function $\Phi$. Then there exists a feature mapping function $g_\theta : V \mapsto \mathbb{R}^M$, such that $K(v_i, v_j) = k_{base}(g_\theta(v_i), g_\theta(v_j))$.*

*Proof.* Let $\Psi$ be the corresponding feature mapping function of the p.s.d. kernel $K$, and then we have $K(v_i, v_j) = \langle \Psi(v_i), \Psi(v_j) \rangle$. Similarly, for $z_i, z_j \in \mathbb{R}^M$, we have $k_{base}(z_i, z_j) = \langle \Phi(z_i), \Phi(z_j) \rangle$. Substitute $g_\theta(v)$ for $z$, and then it is easy to see that $g_\theta(v) = (\Phi^{-1} \circ \Psi)(v)$ is the desired feature mapping function when $\Phi^{-1}$ exists. $\square$

## 4.1 Implementation and Learning Criteria

Theorems 1 and 2 have demonstrated the validity and power of the proposed formulation in Eq. (4). In this section, we discuss how to implement and learn $g_\theta$, $k_{base}$, $s_G$ and $\ell$, respectively.

**Implementation of the Feature Mapping Function** $g_\theta$. The function $g_\theta$ aims to project the feature vector $x_v$ of each node $v$ into a better space for similarity measurement. Our key idea is that in a graph, connected nodes usually share some similar characteristics, and thus changes between nearby nodes in the latent space of nodes should be smooth. Inspired by the concept of kernel smoothing, we consider $g_\theta$ as a *feature smoother* which maps $x_v$ into a *smoothed* latent space according to the graph structure. The kernel smoother estimates a function as the weighted average of neighboring observed data. To be specific, given a node $v \in V$, according to Nadaraya-Watson kernel-weighted average [10], a feature smoothing function is defined as:

$$g(v) = \frac{\sum_{u \in V} k(u, v) p(u)}{\sum_{u \in V} k(u, v)}, \tag{5}$$

where $p$ is a mapping function to compute the feature vector of each node, and here we let $p(v) = x_v$; $k$ is a pre-defined kernel function to capture pairwise relations between nodes. Note that we omit $\theta$ for $g$ here since there are no learnable parameters in Eq. (5). In the context of graphs, the natural choice of computing $k$ is to follow the graph structure, i.e., the structural information within the node's $h$-hop neighborhood.

To compute $g$, we let $\mathbf{A}$ be the adjacent matrix of the given graph $G$ and $\mathbf{I}$ be the identity matrix with the same size. We notice that $\mathbf{I} + \mathbf{D}^{-\frac{1}{2}}\mathbf{A}\mathbf{D}^{-\frac{1}{2}}$ is a valid p.s.d. matrix, where $\mathbf{D}(i,i) = \sum_j \mathbf{A}(i,j)$. Thus we can employ this matrix to define the kernel function $k$. However, in practice, this matrix would lead to numerical instabilities and exploding or vanishing gradients when used for training deep neural networks. To alleviate this problem, we adopt the *renormalization trick* [19]: $\mathbf{I} + \mathbf{D}^{-\frac{1}{2}}\mathbf{A}\mathbf{D}^{-\frac{1}{2}} \rightarrow \bar{\mathbf{A}} = \tilde{\mathbf{D}}^{-\frac{1}{2}}\tilde{\mathbf{A}}\tilde{\mathbf{D}}^{-\frac{1}{2}}$, where $\tilde{\mathbf{A}} = \mathbf{A} + \mathbf{I}$ and $\tilde{\mathbf{D}}(i,i) = \sum_j \tilde{\mathbf{A}}(i,j)$. Then the $h$-hop neighborhood can be computed directly from the $h$ power of $\bar{\mathbf{A}}$, i.e., $\bar{\mathbf{A}}^h$. And the kernel $k$ for node pairs $v_i, v_j \in V$ is computed as $k(v_i, v_j) = \bar{\mathbf{A}}^h(i,j)$. After collecting the feature vector $x_v$ of each node $v \in V$ into a matrix $\mathbf{X}_V$, we rewrite Eq. (5) approximately into its matrix form:

$$g(V) \approx \bar{\mathbf{A}}^h \mathbf{X}_V. \tag{6}$$

Next, we enhance the expressive power of Eq. (6) to model any valid p.s.d. kernels by implementing it with deep neural networks based on the following two aspects. First, we make use of multi-layer perceptrons (MLPs) to model and learn the composite function $\Phi^{-1} \circ \Psi$ in Theorem 2, thanks to the universal approximation theorem [16, 17]. Second, we add learnable weights to different hops of node neighbors. As a result, our final feature mapping function $g_\theta$ is defined as:

$$g_\theta(V) = \left( \sum_h \omega_h \cdot \left( \bar{\mathbf{A}}^h \odot \mathbf{M}^{(h)} \right) \right) \cdot \text{MLP}^{(l)}(\mathbf{X}_V), \tag{7}$$

where $\theta$ means the set of parameters in $g_\theta$; $\omega_h$ is a learnable parameter for the $h$-hop neighborhood of each node $v$; $\odot$ is the Hadamard (element-wise) product; $\mathbf{M}^{(h)}$ is an indicator matrix where $\mathbf{M}^{(h)}(i,j)$ equals to 1 if $v_j$ is a $h$-th hop neighbor of $v_i$ and 0 otherwise. The hyperparameter $l$ controls the number of layers in the MLP.

Equation (7) can be interpreted as a weighted feature aggregation schema around the given node $v$ and its neighbors, which is employed to compute the node representation. It has a close connection with Graph Neural Networks. We leave it in Section 5 for a more detailed discussion.

**Implementation of the Base Kernel $k_{\text{base}}$.** As we have shown in Theorem 2, in order to model an arbitrary p.s.d. kernel, we require that the corresponding feature mapping function $\Phi$ of the base kernel $k_{\text{base}}$ must be invertible, i.e., $\Phi^{-1}$ exists. An obvious choice would let $\Phi$ be an identity function, then $k_{\text{base}}$ will reduce to the dot product between nodes in the latent space. Since $g_\theta$ maps node representations to a *finite* dimensional space, the identity function makes our model directly measure the node similarity in this space. On the other hand, an alternative choice of $k_{\text{base}}$ is the RBF kernel which additionally projects node representations to an *infinite* dimensional latent space before comparison. We compare both implementations in the experiments for further evaluation.

**Data-Driven Similarity Metric $s_G$ and Criteria $\ell$.** In node classification, each node $v_i \in V$ is associated with a class label $y_i \in Y$. We aim to measure node similarity with respect to their class labels other than hand-designed metrics. Naturally, we define the pairwise similarity $s_G$ as:

$$s_G(v_i, v_j) = \begin{cases} 1 & \text{if } y_i = y_j \\ -1 & \text{o/w} \end{cases} \tag{8}$$

However, in practice, it is hard to directly minimize the loss between $K_\theta$ and $s_G$ in Eq. (8). Instead, we consider a "soft" version of $s_G$, where we require that the similarity of node pairs with the same label is greater than those with distinct labels by a margin. Therefore, we train the kernel $K_\theta(v_i, v_j) = k_{\text{base}}(g_\theta(v_i), g_\theta(v_j))$ to minimize the following objective function on *triplets*:

$$\mathcal{L}_K = \sum_{(v_i, v_j, v_k) \in \mathcal{T}} \ell(K_\theta(v_i, v_j), K_\theta(v_i, v_k)), \tag{9}$$

where $\mathcal{T} \subseteq V \times V \times V$ is a set of node triplets: $v_i$ is an anchor, and $v_j$ is a positive of the same class as the anchor while $v_k$ is a negative of a different class. The loss function $\ell$ is defined as:

$$\ell(K_\theta(v_i, v_j), K_\theta(v_i, v_k)) = [K_\theta(v_i, v_k) - K_\theta(v_i, v_j) + \alpha]_+. \tag{10}$$

It ensures that given two positive nodes of the same class and one negative node, the kernel value of the negative should be farther away than the one of the positive by the margin $\alpha$. Here, we present Theorem 3 and its proof to show that minimizing Eq. (9) leads to $K_\theta = s_G$.

**Theorem 3.** *If* $|K_\theta(v_i, v_j)| \leq 1$ *for any* $v_i, v_j \in V$, *minimizing Eq. (9) with* $\alpha = 2$ *yields* $K_\theta = s_G$.

*Proof.* Let $(v_i, v_j, v_k)$ be all triplets satisfying $y_i = y_j$, $y_i \neq y_k$. Suppose that for $\alpha = 2$, Eq. (10) holds for all $(v_i, v_j, v_k)$. It means $K_\theta(v_i, v_k) + 2 \leq K_\theta(v_i, v_j)$ for all $(v_i, v_j, v_k)$. As $|K_\theta(v_i, v_j)| \leq 1$, we have $K_\theta(v_i, v_k) = -1$ for all $(v_i, v_k)$ and $K_\theta(v_i, v_j) = 1$ for all $(v_i, v_j)$. Hence, $K_\theta = s_G$. $\square$

We note that $|K_\theta(v_i, v_j)| \leq 1$ can be simply achieved by letting $k_{\text{base}}$ be the dot product and normalizing all $g_\theta$ to the norm ball. In the following sections, the normalized $K_\theta$ is denoted by $\bar{K}_\theta$.

## 4.2 Inference for Node Classification

Once the kernel function $K_\theta(v_i, v_j) = k_{\text{base}}(g_\theta(v_i), g_\theta(v_j))$ has learned how to measure the similarity between nodes, we can leverage the output of the feature mapping function $g_\theta$ as the node representation for node classification. In this paper, we introduce the following two classifiers.

**Nearest Centroid Classifier.** The nearest centroid classifier extends the $k$-nearest neighbors algorithm by assigning to observations the label of the class of training samples whose centroid is closest to the observation. It does not require additional parameters. To be specific, given a testing node $u$, for all nodes $v_i$ with class label $y_i \in Y$ in the training set, we compute the per-class average similarity between $u$ and $v_i$: $\mu_y = \frac{1}{|V_y|} \sum_{v_i \in V_y} \bar{K}_\theta(u, v_i)$, where $V_y$ is the set of nodes belonging to class $y \in Y$. Then the class assigned to the testing node $u$:

$$y^* = \arg\max_{y \in Y} \mu_y. \tag{11}$$

**Softmax Classifier.** The idea of the softmax classifier is to reuse the ground truth labels of nodes for training the classifier, so that it can be directly employed for inference. To do this, we add the softmax activation $\sigma$ after $g_\theta(v_i)$ to minimize the following objective:

$$\mathcal{L}_Y = -\sum_{v_i \in V} q(y_i) \log(\sigma(g_\theta(v_i))), \tag{12}$$

where $q(y_i)$ is the one-hot ground truth vector. Note that Eq. (12) is optimized together with Eq. (9) in an end-to-end manner. Let $\Psi$ denote the corresponding feature mapping function of $K_\theta$, then we have $K_\theta(v_i, v_j) = \langle \Psi(v_i), \Psi(v_j) \rangle = k_{\text{base}}(g_\theta(v_i), g_\theta(v_j))$. In this case, we use the node feature produced by $\Psi$ for classification since $\Psi$ projects node features into the dot-product space which is a natural metric for similarity comparison. To this end, $k_{\text{base}}$ is fixed to be the identity function for the softmax classifier, so that we have $\langle \Psi(v_i), \Psi(v_j) \rangle = \langle g_\theta(v_i), g_\theta(v_j) \rangle$ and thus $\Psi(v_i) = g_\theta(v_i)$.

## 5 Discussion

Our feature mapping function $g_\theta$ proposed in Eq. (7) has a close connection with Graph Convolutional Networks (GCNs) [19] in the way of capturing node latent representations. In GCNs and most of their variants, each layer leverages the following aggregation rule:

$$\mathbf{H}^{(l+1)} = \rho\left(\bar{\mathbf{A}} \mathbf{H}^{(l)} \mathbf{W}^{(l)}\right), \tag{13}$$

where $\mathbf{W}^{(l)}$ is a layer-specific trainable weighting matrix; $\rho$ denotes an activation function; $\mathbf{H}^{(l)} \in \mathbb{R}^{N \times D}$ denotes the node features in the $l$-th layer, and $\mathbf{H}^0 = \mathbf{X}$. Through stacking multiple layers, GCNs aggregate the features for each node from its $L$-hop neighbors recursively, where $L$ is the network depth. Compared with the proposed $g_\theta$, GCNs actually interleave two basic operations of $g_\theta$: feature transformation and Nadaraya-Watson kernel-weighted average, and repeat them recursively.

We contrast our approach with GCNs in terms of the following aspects. First, our aggregation function is derived from the kernel perspective, which is novel. Second, we show that aggregating features in a recursive manner is inessential. Powerful $h$-hop node representations can be obtained by our model where aggregation is performed only once. As a result, our approach is more efficient both in storage and time when handling very large graphs, since no intermediate states of the network have to be kept. Third, our model is flexible and complementary to GCNs: our function $g_\theta$ can be directly replaced by GCNs and other variants, which can be exploited for future work.

**Time and Space Complexity.** We assume the number of features $F$ is fixed for all layers and both GCNs and our method have $L \geq 2$ layers. We count matrix multiplications as in [7]. GCN's time complexity is $\mathcal{O}(L\|\bar{\mathbf{A}}\|_0 F + L|V|F^2)$, where $\|\bar{\mathbf{A}}\|_0$ is the number of nonzeros of $\bar{\mathbf{A}}$ and $|V|$ is the number of nodes in the graph. While ours is $\mathcal{O}(\|\bar{\mathbf{A}}^h\|_0 F + L|V|F^2)$, since we do not aggregate features recursively. Obviously, $\|\bar{\mathbf{A}}^h\|_0$ is constant but $L\|\bar{\mathbf{A}}\|_0$ is linear to $L$. For space complexity, GCNs have to store all the feature matrices for recursive aggregation which needs $\mathcal{O}(L|V|F + LF^2)$ space, where $LF^2$ is for storing trainable parameters of all layers, and thus the first term is linear to $L$. Instead, ours is $\mathcal{O}(|V|F + LF^2)$ where the first term is again constant to $L$. Our experiments indicate that we save 20% (0.3 ms) time and 15% space on Cora dataset [22] than GCNs.

## 6 Experiments

We evaluate the proposed kernel-based approach on three benchmark datasets: Cora [22], Citeseer [11] and Pubmed [30]. They are citation networks, where the task of node classification is to classify academic papers of the network (graph) into different subjects. These datasets contain bag-of-words features for each document (node) and citation links between documents.

We compare our approach to five state-of-the-art methods: GCN [19], GAT [38], FastGCN [5], JK [41] and KLED [9]. KLED is a kernel-based method, while the others are based on deep neural networks. We test all methods in the supervised learning scenario, where all data in the training set are used for training. We evaluate the proposed method in two different experimental settings according to FastGCN [5] and JK [41], respectively. The statistics of the datasets together with their data split settings (i.e., the number of samples contained in the training, validation and testing sets, respectively) are summarized in Table 1. Note that there are more training samples in the data split of JK [41] than FastGCN [5]. We report the average means and standard deviations of node classification accuracy which are computed from ten runs as the evaluation metrics.

Table 1: Overview of the three evaluation datasets under two different data split settings.

| Dataset | Nodes | Edges | Classes | Features | Data split of FastGCN [5] | Data split of JK [41] |
|---|---|---|---|---|---|---|
| Cora [22] | 2,708 | 5,429 | 7 | 1,433 | 1,208 / 500 / 1,000 | 1,624 / 542 / 542 |
| Citeseer [11] | 3,327 | 4,732 | 6 | 3,703 | 1,827 / 500 / 1,000 | 1,997 / 665 / 665 |
| Pubmed [30] | 19,717 | 44,338 | 3 | 500 | 18,217 / 500 / 1,000 | - |

### 6.1 Variants of the Proposed Method

As we have shown in Section 4.1, there are alternative choices to implement each component of our framework. In this section, we summarize all the variants of our method employed for evaluation.

**Choices of the Feature Mapping Function** $g$. We implement the feature mapping function $g_\theta$ according to Eq. (7). In addition, we also choose GCN and GAT as the alternative implementations of $g_\theta$ for comparison, and denote them by $g_{\text{GCN}}$ and $g_{\text{GAT}}$, respectively.

**Choices of the Base Kernel** $k_{\text{base}}$. The base kernel $k_{\text{base}}$ has two different implementations: the dot product which is denoted by $k_{\langle \cdot, \cdot \rangle}$, and the RBF kernel which is denoted by $k_{\text{RBF}}$. Note that when the softmax classifier is employed, we set the base kernel to be $k_{\langle \cdot, \cdot \rangle}$.

**Choices of the Loss** $\mathcal{L}$ **and Classifier** $\mathcal{C}$. We consider the following three combinations of the loss function and classifier. (1) $\mathcal{L}_K$ in Eq. (9) is optimized, and the nearest-centroid classifier $\mathcal{C}_K$ is employed for classification. This combination aims to evaluate the effectiveness of the learned kernel. (2) $\mathcal{L}_Y$ in Eq. (12) is optimized, and the softmax classifier $\mathcal{C}_Y$ is employed for classification. This combination is used in a baseline without kernel methods. (3) Both Eq. (9) and Eq. (12) are optimized, and we denote this loss by $\mathcal{L}_{K+Y}$. The softmax classifier $\mathcal{C}_Y$ is employed for classification. This combination aims to evaluate how the learned kernel improves the baseline method.

In the experiments, we use $\mathcal{K}$ to denote kernel-based variants and $\mathcal{N}$ to denote ones without the kernel function. All these variants are implemented by MLPs with two layers. Due to the space limitation, we ask the readers to refer to the supplementary material for implementation details.

## 6.2 Results of Node Classification

The means and standard deviations of node classification accuracy (%) following the setting of FastGCN [5] are organized in Table 2. Our variant of $\mathcal{K}_3$ sets the new state of the art on all datasets. And on Pubmed dataset, all our variants improve previous methods by a large margin. It proves the effectiveness of employing kernel methods for node classification, especially on datasets with large graphs. Interestingly, our non-kernel baseline $\mathcal{N}_1$ even achieves the state-of-the-art performance, which shows that our feature mapping function can capture more flexible structural information than previous GCN-based approaches. For the choice of the base kernel, we can find that $\mathcal{K}_2$ outperforms $\mathcal{K}_1$ on two large datasets: Citeseer and Pubmed. We conjecture that when handling complex datasets, the non-linear kernel, e.g., the RBF kernel, is a better choice than the liner kernel.

To evaluate the performance of our feature mapping function, we report the results of two variants $\mathcal{K}_1^*$ and $\mathcal{K}_2^*$ in Table 2. They utilize GCN and GAT as the feature mapping function respectively. As expected, our $\mathcal{K}_1$ outperforms $\mathcal{K}_1^*$ and $\mathcal{K}_2^*$ among most datasets. This demonstrates that the recursive aggregation schema of GCNs is inessential, since the proposed $g_\theta$ aggregates features only in a single step, which is still powerful enough for node classification. On the other hand, it is also observed that both $\mathcal{K}_1^*$ and $\mathcal{K}_2^*$ outperform their original non-kernel based implementations, which shows that learning with kernels yields better node representations.

Table 3 shows the results following the setting of JK [41]. Note that we do not evaluate on Pubmed in this setup since its corresponding data split for training and evaluation is not provided by [41]. As expected, our method achieves the best performance among all datasets, which is consistent with the results in Table 2. For Cora, the improvement of our method is not so significant. We conjecture that the results in Table 3 involve more training data due to different data splits, which narrows the performance gap between different methods on datasets with small graphs, such as Cora.

Table 2: Accuracy (%) of node classification following the setting of FastGCN [5].

| Method | Cora [22] | Citeseer [11] | Pubmed [30] |
|---|---|---|---|
| KLED [9] | 82.3 | - | 82.3 |
| GCN [19] | 86.0 | 77.2 | 86.5 |
| GAT [38] | 85.6 | 76.9 | 86.2 |
| FastGCN [5] | 85.0 | 77.6 | 88.0 |
| $\mathcal{K}_1 = \{k_{\langle \cdot, \cdot \rangle}, g_\theta, \mathcal{L}_K, \mathcal{C}_K\}$ | $86.68 \pm 0.17$ | $77.92 \pm 0.25$ | $89.22 \pm 0.17$ |
| $\mathcal{K}_2 = \{k_{\text{RBF}}, g_\theta, \mathcal{L}_K, \mathcal{C}_K\}$ | $86.12 \pm 0.05$ | $78.68 \pm 0.38$ | $89.36 \pm 0.21$ |
| $\mathcal{K}_3 = \{k_{\langle \cdot, \cdot \rangle}, g_\theta, \mathcal{L}_{K+Y}, \mathcal{C}_Y\}$ | $\mathbf{88.40 \pm 0.24}$ | $\mathbf{80.28 \pm 0.03}$ | $\mathbf{89.42 \pm 0.01}$ |
| $\mathcal{N}_1 = \{g_\theta, \mathcal{L}_Y, \mathcal{C}_Y\}$ | $87.56 \pm 0.14$ | $79.80 \pm 0.03$ | $89.24 \pm 0.14$ |
| $\mathcal{K}_1^* = \{k_{\langle \cdot, \cdot \rangle}, g_{\text{GCN}}, \mathcal{L}_K, \mathcal{C}_K\}$ | $87.04 \pm 0.09$ | $77.12 \pm 0.23$ | $87.84 \pm 0.12$ |
| $\mathcal{K}_2^* = \{k_{\langle \cdot, \cdot \rangle}, g_{\text{GAT}}, \mathcal{L}_K, \mathcal{C}_K\}$ | $86.10 \pm 0.33$ | $77.92 \pm 0.19$ | - |

Table 3: Accuracy (%) of node classification following the setting of JK [41].

| Method | Cora [22] | Citeseer [11] |
|---|---|---|
| GCN [19] | $88.20 \pm 0.70$ | $77.30 \pm 1.30$ |
| GAT [38] | $87.70 \pm 0.30$ | $76.20 \pm 0.80$ |
| JK-Concat [41] | $89.10 \pm 1.10$ | $78.30 \pm 0.80$ |
| $\mathcal{K}_3 = \{k_{\langle \cdot, \cdot \rangle}, g_\theta, \mathcal{L}_{K+Y}, \mathcal{C}_Y\}$ | $\mathbf{89.24 \pm 0.31}$ | $\mathbf{80.78 \pm 0.28}$ |

## 6.3 Ablation Study on Node Feature Aggregation Schema

In Table 4, we implement three variants of $\mathcal{K}_3$ (2-hop and 2-layer with $\omega_h$ by default) to evaluate the proposed node feature aggregation schema. We answer the following three questions. (1) *How does performance change with fewer (or more) hops?* We change the number of hops from 1 to 3, and the performance improves if it is larger, which shows capturing long-range structures of nodes is important. (2) *How many layers of MLP are needed?* We show results with different layers ranging from 1 to 3. The best performance is obtained with two layers, while networks overfit the data when

more layers are employed. (3) *Is it necessary to have a trainable parameter $\omega_h$?* We replace $\omega_h$ with a fixed constant $c^h$, where $c \in (0, 1]$. We can see larger $c$ improves the performance. However, all results are worse than learning a weighting parameter $\omega_h$, which shows the importance of it.

Table 4: Results of accuracy (%) with different settings of the aggregation schema.

| Variants of $\mathcal{K}_3$ | Cora [22] | Citeseer [11] | Pubmed [30] |
|---|---|---|---|
| Default | **88.40 $\pm$ 0.24** | **80.28 $\pm$ 0.03** | 89.42 $\pm$ 0.01 |
| 1-hop | 85.56 $\pm$ 0.02 | 77.73 $\pm$ 0.02 | 88.98 $\pm$ 0.01 |
| 3-hop | 88.25 $\pm$ 0.01 | 80.13 $\pm$ 0.01 | **89.53 $\pm$ 0.01** |
| 1-layer | 82.60 $\pm$ 0.01 | 77.63 $\pm$ 0.01 | 85.80 $\pm$ 0.01 |
| 3-layer | 86.33 $\pm$ 0.04 | 78.53 $\pm$ 0.20 | 89.46 $\pm$ 0.05 |
| $c = 0.25$ | 69.33 $\pm$ 0.09 | 74.48 $\pm$ 0.03 | 84.68 $\pm$ 0.02 |
| $c = 0.50$ | 76.98 $\pm$ 0.10 | 77.47 $\pm$ 0.04 | 86.45 $\pm$ 0.01 |
| $c = 0.75$ | 84.25 $\pm$ 0.01 | 77.99 $\pm$ 0.01 | 87.45 $\pm$ 0.01 |
| $c = 1.00$ | 87.31 $\pm$ 0.01 | 78.57 $\pm$ 0.01 | 88.68 $\pm$ 0.01 |

### 6.4 t-SNE Visualization of Node Embeddings

We visualize the node embeddings of GCN, GAT and our method on Citeseer with t-SNE. For our method, we use the embedding of $\mathcal{K}_3$ which obtains the best performance. Figure 2 illustrates the results. Compared with other methods, our method produces a more compact clustering result. Specifically our method clusters the "red" points tightly, while in the results of GCN and GAT, they are loosely scattered into other clusters. This is caused by the fact that both GCN and GAT minimize the classification loss $\mathcal{L}_Y$, only targeting at accuracy. They tend to learn node embeddings driven by those classes with the majority of nodes. In contrast, $\mathcal{K}_3$ are trained with both $\mathcal{L}_K$ and $\mathcal{L}_Y$. Our kernel-based similarity loss $\mathcal{L}_K$ encourages data within the same class to be close to each other. As a result, the learned feature mapping function $g_\theta$ encourages geometrically compact clusters.

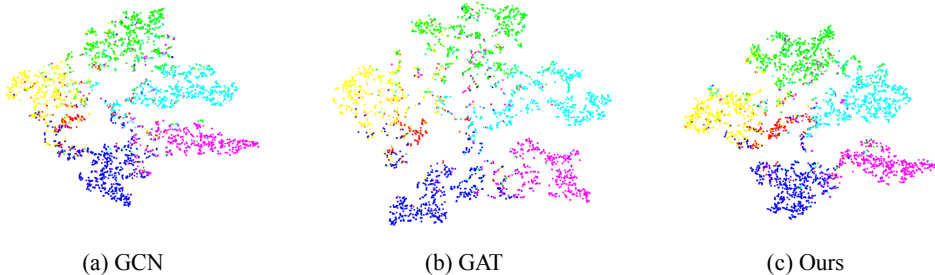

(a) GCN                    (b) GAT                    (c) Ours

Figure 2: t-SNE visualization of node embeddings on Citeseer dataset.

Due to the space limitation, we ask the readers to refer to the supplementary material for more experiment results, such as the results of link prediction and visualization on other datasets.

## 7 Conclusions

In this paper, we introduce a kernel-based framework for node classification. Motivated by the design of graph kernels, we learn the kernel from ground truth labels by decoupling the kernel function into a base kernel and a learnable feature mapping function. More importantly, we show that our formulation is valid as well as powerful enough to express any p.s.d. kernels. Then the implementation of each component in our approach is extensively discussed. From the perspective of kernel smoothing, we also derive a novel feature mapping function to aggregate features from a node's neighborhood. Furthermore, we show that our formulation is closely connected with GCNs but more powerful. Experiments on standard node classification benchmarks are conducted to evaluated our approach. The results show that our method outperforms the state of the art.

**Acknowledgments**

This work is funded by ARO-MURI-68985NSMUR and NSF 1763523, 1747778, 1733843, 1703883.

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
