[Supplementary Material · 6235_supp.pdf]

# Rethinking Kernel Methods for Node Representation Learning on Graphs: Supplementary Material

**Yu Tian**\*
Rutgers University
yt219@cs.rutgers.edu

**Long Zhao**\*
Rutgers University
lz311@cs.rutgers.edu

**Xi Peng**
University of Delaware
xipeng@udel.edu

**Dimitris N. Metaxas**
Rutgers University
dnm@cs.rutgers.edu

## 1 Implementation Details

We use different network settings for the combinations of the loss function and inference method in Section 6.1 of the original paper. For Variant (1), we choose the output dimension of the first and second layers to be 512 and 128, respectively. We train this combination with 10 epochs on Cora and Citeseer and 100 epochs on Pubmed.

For GAT [4], due to its large memory cost, its output dimension of the first and second layers is chosen to be 64 and 8, respectively.

For Variants (2) and (3), the output dimension of the first layer is chosen to be 16. The output dimension of the second layer is the same as the number of node classes. We train this combination 100 epochs for GAT and 200 epochs for other setups.

In Eq. (9) of the original paper, we randomly sample 10,000 triplets in each epoch. In Eq. (10) of the original paper, $\alpha$ is set to be 0.1 for all datasets. All methods are optimized using Adam [1] with the learning rate of 0.01. We use the best model achieved on the validation set for testing. Each result is reported based on an average over 10 runs.

## 2 Additional Experimental Results

### 2.1 Results of Link Prediction

In addition to node classification, we also conduct experiments for link prediction to demonstrate the generalizability of the proposed framework in different graph-based tasks. We train the models using an *incomplete* version of the three citation datasets (Cora, Citeseer and Pubmed) according to [2]: the node features remain but parts of the citation links (edges) are missing. The validation and test sets are constructed following the setup of [2].

We choose $k_{\text{base}}$ to be the dot product and set $g_\theta$ to be the feature mapping function. Given graph $G = (V, E)$, for $v_i, v_j \in V$, the similarity measure is defined as:

$$s_G(v_i, v_j) = \begin{cases} 1 & \text{if } (v_i, v_j) \in E \\ 0 & \text{o/w} \end{cases} \tag{1}$$

The feature mapping function $g_\theta$ can be learned by minimizing the following objective function in a data-driven manner:

$$\mathcal{L}_K = \sum_{(v_i, v_j) \in \mathcal{D}} \ell(K_\theta(v_i, v_j), s_G(v_i, v_j)), \tag{2}$$

where $\mathcal{D}$ is the set of training edges, and $\ell$ is the binary cross entropy loss.

Table 1 summarizes the link prediction results of our kernel-based method, the variational graph autoencoder (VGAE) [2] and its non-probabilistic variant (GAE). Our kernel-based method is highly comparable with these state-of-the-art methods, showing the potential of applying the proposed framework in different applications on graphs.

Table 1: Accuracy (%) of link prediction.

| Method | Cora | | Citeseer | | Pubmed | |
|---|---|---|---|---|---|---|
| | AUC | AP | AUC | AP | AUC | AP |
| GAE [2] | $91.0 \pm 0.02$ | $92.0 \pm 0.03$ | $89.5 \pm 0.04$ | $89.9 \pm 0.05$ | $\mathbf{96.4 \pm 0.00}$ | $\mathbf{96.5 \pm 0.00}$ |
| VGAE [2] | $91.4 \pm 0.01$ | $92.6 \pm 0.01$ | $90.8 \pm 0.02$ | $\mathbf{92.0 \pm 0.02}$ | $94.4 \pm 0.02$ | $94.7 \pm 0.02$ |
| Ours | $\mathbf{93.1 \pm 0.06}$ | $\mathbf{93.2 \pm 0.07}$ | $\mathbf{90.9 \pm 0.08}$ | $91.8 \pm 0.04$ | $94.5 \pm 0.03$ | $94.2 \pm 0.01$ |

## 2.2 t-SNE visualization on Cora

We visualize the node embeddings of GCN [3], GAT [4] and our method on Cora with t-SNE in Fig. 1. Our method produces tight and clear clustering embeddings (especially for the "red" points and "violet" points), which shows that compared with GCN and GAT, our method is able to learn more reasonable feature embeddings for nodes.

(a) GCN                    (b) GAT                    (c) Ours

Figure 1: t-SNE visualization of node embeddings on Cora dataset.

## Footnotes

\*indicates equal contributions.