[Reviews · NeurIPS 2019]

Reviewer 1



After rebuttal: thank you for the additional experiments. They strengthen the empirical contribution of the paper, so I've increased my score to a 7. ________________ Originality: The paper is a novel combination of known techniques: by reinterpreting the the iterative node aggregation procedure of Kipf et al's GCN as feature smoothing technique, they develop a novel feature mapping function for learning positive semi-definite (psd) graph kernels. The key difference from the Kipf et al approach is they separate the node aggregation and non-linear representation learning components: node features are the output of a multi-layer perceptron and then aggregated once (rather than at every layer) by a multi-hop aggregation function. They argue theoretically that this approach is universal in the sense that it can approximate any invertible psd kernel. Quality: I thought the empirical results of the paper were interesting because they suggest that decoupling the aggregation and representation learning components of GCN-style models leads to better performance (at least on these datasets). I would have liked to see more ablation experiments to explore this further: e.g. how does performance change with fewer / more hops? What role is \omega_h playing empirically? e.g. Would a fixed constant, say c^h, c \in [0, 1] lead to similar performance or is it necessary to have a trainable parameter? Do you need an MLP or would a single projection layer suffice? If not, how many layers do you need? I didn't find the theory as interesting - it essentially says you can learn psd kernels (Theorem 1) & that they are universal (Theorem 2). That may be true, but I'm not sure it buys us anything beyond the fact that you're learning a similarity function? Clarity: in general the paper was well-written and clear. That said, table 1 was missing what units it was in - accuracy? AUC? Also are those standard error or standard deviations? Significance: I thought this is an interesting contribution because it gives a new perspective on the relative importance of the components of GCN-style models. That said (as mentioned above), a proper ablation study that explored the implications of this perspective would make this a far stronger contribution.

Reviewer 2



The paper presents a kernel (and MLP)-based approach for node classification in a graph. The system is rather complex and takes more than one reading to understand it. The paper lists four contributions, two more than the ones I reported in the "Contributions" section. I do not fully agree with the remaining ones: "we theoretically show that our formulation is powerful enough to express any valid positive semidefinite kernels.". As the authors state, their results assume the feature mapping \phi to be invertible, which I do not think it is true for any \phi. The authors should smooth the claim or prove any kernel is invertible. I understand they use a MLP to approximate \phi^-1, but that's an approximation, it should be proved how precise that is for every kernel for the claim to be true in its current form. "our model captures the structural information of a node by aggregation in a single step, other than a recursive manner, thus is more efficient". However, an analysis of the efficiency of the method, together with a comparison with related methods, is missing in the paper. If I understand correctly the statement in rows 114-115, it should be rephrased to "it is impossible to..." (see [1] for a justification). The inline equation in row 119 reminds me of the Mapping Kernels framework, it would be fair to assess whether I am correct and, in case, cite the paper [2]. Theorem 1 is straightforward, there is no need for a Theorem. It is not easy to understand how significant is the improvement, because a description of the datasets is missing in the paper and no statistical analysis is provided. However, I see the main contribution of the paper in the novel theoretically-guided approach. ---------------- [1] Ramon, J., & Gärtner, T. (2003). Expressivity versus Efficiency of Graph Kernels. Proceedings of the First International Workshop on Mining Graphs, Trees and Sequences {(MGTS-2003)}, 65–74. [2] Shin, K., & Kuboyama, T. (2008). A generalization of Haussler’s convolution kernel: mapping kernel. In Proceedings of the 25th international conference on Machine learning (pp. 944–951). Helsinki, Finland: ACM. https://doi.org/10.1145/1390156.1390275

Reviewer 3



This paper proposed a learnable kernel-based framework for node classification. To efficiently learn the kernel, the paper proposed a mechanism for node feature aggregation and a data-driven similarity metric employed during the training phase. The problem and solution are both interesting, however the experiments are conducted on three datasets. What's the time and space complexity of this method?

[Author Response · NeurIPS 2019]

We sincerely thank the reviewers for their positive feedback. We appreciate they find our contribution **"interesting"**
(R1, R3) and **"novel"** (R1, R2); the approach **"gives a new perspective"** (R1) and is **"novel theoretically-guided"**
(R2, R3); the empirical results are **"interesting"** (R1). We also appreciate they find the paper is **"well-written"** and
**"clear"** (R1). We address the main concerns raised by the reviewers in the rebuttal, and will incorporate all suggestions
for changes in the camera-ready version. We sincerely hope this will help the reviewers to finalize their judgments.

**Q1. Ablation study on node feature aggregation schema. (R1)**

In Table 1, we implement three variants of $\mathcal{K}_3$ (2-hop and 2-layer
with $\omega_h$ by default) to answer the following three questions. We
report the mean accuracy following the setting of FastGCN.

*(1) How does performance change with fewer (or more) hops?*
We change the number of hops from 1 to 3, and the performance
improves if it is larger, which shows capturing long-range structures
of nodes is important. *(2) How many layers of MLP do you need?*
We show results with different layers ranging from 1 to 3. The best
performance is obtained with 2 layers, while networks overfit the
data when more layers are employed. *(3) Is it necessary to have*
*a trainable parameter $\omega_h$?* We replace $\omega_h$ with a fixed constant $c^h$,
where $c \in (0, 1]$. We can see larger $c$ improves the performance.
However, all results are worse than learning a weighting parameter
$\omega_h$, which shows the importance of it.

Table 1: Results of accuracy (%).

| Variants of $\mathcal{K}_3$ | Cora | Citeseer | Pubmed |
|---|---|---|---|
| Default | **88.40** | **80.28** | 89.42 |
| 1-hop | 85.56 | 77.73 | 88.98 |
| 3-hop | 88.25 | 80.13 | **89.53** |
| 1-layer | 82.60 | 77.63 | 85.80 |
| 3-layer | 86.33 | 78.53 | 89.46 |
| $c = 0.25$ | 69.33 | 74.48 | 84.68 |
| $c = 0.50$ | 76.98 | 77.47 | 86.45 |
| $c = 0.75$ | 84.25 | 77.99 | 87.45 |
| $c = 1.00$ | 87.31 | 78.57 | 88.68 |

**Q2. Time and space complexity of the proposed approach compared with GCNs. (R2, R3)**

We assume the number of features $F$ is fixed for all layers and each method has $L \geq 2$ layers. *(1) Time complexity.* We
count matrix multiplications as in [1]. GCN's time complexity is $\mathcal{O}(L\|\bar{\mathbf{A}}\|_0 F + L|V|F^2)$, where $\|\bar{\mathbf{A}}\|_0$ is the number
of nonzeros of $\bar{\mathbf{A}}$ and $|V|$ is the number of nodes in the graph. While ours is $\mathcal{O}(\|\bar{\mathbf{A}}^h\|_0 F + L|V|F^2)$, since we do not
aggregate features recursively. Obviously, $\|\bar{\mathbf{A}}^h\|_0$ is constant but $L\|\bar{\mathbf{A}}\|_0$ is linear to $L$. *(2) Space complexity.* GCNs
have to store all the feature matrices for recursive aggregation which needs $\mathcal{O}(L|V|F + LF^2)$ space, and thus the
first term is linear to $L$. Instead, ours is $\mathcal{O}(|V|F + LF^2)$ where the first term is again constant to $L$. Our experiments
indicate that we save **20% (0.3 ms) time** and **15% space** on Cora dataset than GCNs.

**Q3. Justification on the necessity and clarity of the theorems. (R1, R2)**

*(1) Why Theorems 1 & 2 are necessary?* Theorem 1 demonstrates the validity while Theorem 2 shows the power of our
approach. They theoretically identify the upper bound of our expressive power, which is crucial in guaranteeing the
general property when applying our approach to different tasks. *(2) Is our formulation powerful enough to express any*
*p.s.d. kernels?* We do not require every $\Phi$ of $k_{base}$ to be invertible. Theorem 2 says given a base kernel $k_{base}$ (with
invertible $\Phi$), we can express any valid p.s.d kernel $K$ with a powerful $g_\theta$. Thus Theorem 2 is correct given we can find
at least one invertible base kernel: one choice is $k_{base}(z_i, z_j) = \langle z_i, z_j \rangle$, where $\Phi$ is the identity function. But we agree
that, in practise, MLP may not express such a powerful $g_\theta$. We have discussed this fact in the main paper (line 178-185)
and we will make it more clear in the final version.

**Q4. Clarification on datasets and empirical results. (R1, R2, R3)**

*(1) Clarification on datasets.* We have provided detailed description of the datasets in the supplementary materials
which can be moved to the main paper as suggested by R2 to make it more clear. Following the **standard protocols**
in the literature, we evaluated our approach on **three** different tasks under **two** settings (following FastGCN and JK,
respectively). Tables 1 and 2 show the means and standard deviations of totally **ten runs**. We believe the current
results are sufficient to demonstrate our superior performance. We also plan to apply our method to other real-world
problems in the future work. *(2) Significance of empirical results.* We perform **t-test** on Table 2 and obtain $p$-**values**
**less than 0.01** with all other methods except the one with JK on Cora (0.77). This proves the statistical significance of
our improvements compared with the state of the art. For the one with 0.77, we argue that it uses more training data on
a relatively small graph, which narrows the performance gap.

At last, we will reformulate the corresponding parts and cite related references as suggested (R2). The remaining
questions about writing will be carefully addressed. We thank the reviewers for their careful feedback and consideration.

# References

[1] Chiang *et al.* Cluster-GCN: An Efficient Algorithm for Training Deep and Large Graph Convolutional Networks. In *KDD*, 2019.


[Meta-Review · NeurIPS 2019]

The reviewers were initially positive about the paper but not confident about their opinions. After the author response, a few of their questions were answered, and the discussion led to a consensus that the paper should be a solid contribution to the NeurIPS program.